# Time series analyses based on the joint lagged effect analysis of pollution and meteorological factors of hemorrhagic fever with renal syndrome and the construction of prediction model

**Ye Chen[1☯], Weiming Hou[2☯], Jing Dong [2]***

**1** Department of Infectious Disease, Shenyang Center for Disease Control and Prevention, Shenyang, PR China, **2** Department of Occupational and Environmental Health, School of Public Health, China Medical University, Shenyang, Peoples' Republic of China

☯ These authors contributed equally to this work.

\* jdong@cmu.edu.cn

## Abstract

### Background

Hemorrhagic fever with renal syndrome (HFRS) is a rodent-related zoonotic disease induced by hantavirus. Previous studies have identified the influence of meteorological factors on the onset of HFRS, but few studies have focused on the stratified analysis of the lagged effects and interactions of pollution and meteorological factors on HFRS.

### Methods

We collected meteorological, contaminant and epidemiological data on cases of HFRS in Shenyang from 2005–2019. A seasonal autoregressive integrated moving average (SARIMA) model was used to predict the incidence of HFRS and compared with Holt-Winters three-parameter exponential smoothing model. A distributed lag nonlinear model (DLNM) with a maximum lag period of 16 days was applied to assess the lag, stratification and extreme effects of pollution and meteorological factors on HFRS cases, followed by a generalized additive model (GAM) to explore the interaction of $SO_2$ and two other meteorological factors on HFRS cases.

### Results

The SARIMA monthly model has better fit and forecasting power than its own quarterly model and the Holt-Winters model, with an optimal model of $(1,1,0)(2,1,0)_{12}$. Overall, environmental factors including humidity, wind speed and $SO_2$ were correlated with the onset of HFRS and there was a non-linear exposure-lag-response association. Extremely high $SO_2$ increased the risk of HFRS incidence, with the maximum RR values: 2.583 (95% CI:1.145,5.827). Extremely low windy and low $SO_2$ played a significant protective role on HFRS infection, with the minimum RR values: 0.487 (95%CI:0.260,0.912) and 0.577 (95% CI:0.370,0.898), respectively. Interaction indicated that the risk of HFRS infection reached its highest when increasing daily $SO_2$ and decreasing humidity.

**Data Availability Statement:** Patient data are protected by the Shenyang CDC and are unsuitable for public sharing. The HFRS data is not allowed to

be publicly shared due to local infection disease law. Interested parties can apply for the data by contacting the Ethics Committee of Shenyang Center for Disease Control and Prevention (email address: 749278935@qq.com).

**Funding:** The author(s) received no specific funding for this work.

**Competing interests:** The authors have declared that no competing interests exist.

## Conclusions

The SARIMA model may help to enhance the forecast of monthly HFRS incidence based on a long-range dataset. Our study had shown that environmental factors such as humidity and $SO_2$ have a delayed effect on the occurrence of HFRS and that the effect of humidity can be influenced by $SO_2$ and wind speed. Public health professionals should take greater care in controlling HFRS in low humidity, low windy conditions and 2–3 days after $SO_2$ levels above 200 μg/m$^3$.

## Author summary

China has the highest number of people infected with hemorrhagic fever with renal syndrome (HFRS) in the world, and Shenyang, located in the northeast, is a high prevalence area for infection in China. Previous studies have found that there are several analytical methods on outbreak prediction and that HFRS infection is climate-related. However, HFRS has been less studied in terms of comparative time series prediction, and the link between outbreaks and atmospheric pollution and the identification of the joint effects of meteorological factors affecting this link have not been studied. These are the two main focuses of this study. A synchronous periodicity and seasonality between pollutants, climate change and HFRS infection were found throughout the study area, both located in spring-summer and winter-related. Specifically, on the one hand, high sulfur dioxide concentrations increase the risk of developing HFRS. On the other hand, the combined effect of climate and pollutants on HFRS became increasingly sensitive over time, showing as the highest risk of contracting HFRS when increasing daily sulfur dioxide and decreasing humidity. Time series analysis showed that seasonal SARIMA models are more suitable for prediction, and the association between climate and pollution and HFRS infection has been confirmed within the time series analysis. The above findings help to improve the understanding of the transmission effects of HFRS in different meteorological and pollution levels and the prediction of HFRS outbreak epidemics.

## 1. Introduction

This hantavirus-like infection has attracted world attention during the Korean War since it was first described in Chinese texts 900 years ago [1]. Two clinical syndromes caused by hantavirus infection have been characterized: hantavirus cardiopulmonary syndrome(HCPS), prevalent mainly in America, and Hemorrhagic fever with renal syndrome(HFRS), found in Eurasia [2]. HFRS [1], characterized by headache, fever, back pain, abdominal pain and acute renal insufficiency [3], has caused a variety of public problems, with 30,000–60,000 cases per year in mainland China in the 1990s [4]. In Europe, over 9000 cases of HFRS are reported annually, and most cases associated with HFRS are diagnosed in parts of Europe in Russia, Finland and Sweden [5,6]. China has the highest rate of HFRS disease in the world, with domestic HFRS cases constituting 90% of the total number of cases worldwide each year [7,8]. HFRS is a zoonotic disease associated with rodents and a legally reported disease in China [9].

In the past, descriptive statistical methods were mostly used for the analysis of infectious diseases such as HFRS. Nonetheless, effective prediction of short and medium term HFRS incidence can provide a reliable basis for the Center of Disease Control (CDC), as well as scientific

theory and support for national infectious disease prevention and control planning [10]. With the application of big data prediction technology in various fields, it also supplies approaches for the development of HFRS prediction technology. At present, there are many ways to study data prediction technology at home and abroad, and they have been broadly applied good results [11]. Regarding infectious disease forecasting, autoregressive integrated moving average (ARIMA) and Holt-Winters models are one of the most representative and widely used models in time series forecasting [12]. Today, predictive modeling has been used in many studies in epidemiological research. Some researchers applied the seasonal autoregressive fractionally integrated moving average (SARFIMA) model to predict renal syndrome hemorrhagic fever [13]. Furthermore, a European study had applied mathematical modeling to explore the pathogenesis and impact of influenza and pathogens [14]. Junyu He et al. applied prophet and ARIMA models in 2022 to evaluate the predictive effect of HFRS incidence in the Chinese region from 1950–2018 [15].

China began to show a declining trend in HFRS cases in the early 1990s and the annual number has fallen more significantly since 2000, from 37,814 cases in 2000 to 11,248 cases in 2007 [4,16]. Seven hantavirus serotypes/genotypes have been identified in China [17] Of these, Hantavirus and Seoul virus are the main pathogens of HFRS, and cases caused by Hantavirus account for about 70% of domestic HFRS cases [18]. Climate and environmental changes might impact the reservoir ecology and dynamics of rodent carriers, thereby triggering the spread of hantavirus transmission [4,19].

Climatic conditions are broadly regarded as some of the most pivotal factors affecting rodent population dynamics and contributing to more cases of HFRS in humans [20]. Some studies have found a correlation between the climatic factors and HFRS. For instance, a systematic evaluation of climate variability and human hantavirus infection in Europe was previously carried out by J. Roda Gracia et al. In 2010, some researches in China used time-series Poisson regression model to examine the independent effect of climate variables on the spread of HFRS, pointing out the important role of climate variation in the transmission of HFRS in northeastern China [21].It will contribute to future international discussions on zoonotic diseases in the context of climate change [22]. Yizhe Luo et al. found that temperature with a lag of 6 months (RR = 3.05) and precipitation with a lag of 0 months (RR = 2.08) had the greatest effect on the incidence of HFRS [23]. Also, recent Chinese studies have shown that temperature and relative humidity have an approximately parabolic or linear effect on the incidence of HFRS in 2022 [24]. Yet little research has been done on the correlation between pollutants and HFRS epidemics. And few studies have synthesized the lagged effects of diverse environmental variables on the onset of HFRS and analyzed the interactions among them. Therefore, we explored the relationship between meteorological and pollutant factors and the onset of HFRS, and speculated that the interaction among environmental factors is of attention for HFRS.

In this study, the incidence rate was predicted by comparison using time series analysis using HFRS surveillance data in Shenyang, followed by the application of boosted regression trees (BRT) verifying the fit and interaction among the environmental factors, and the lag effect and interaction of meteorological and pollutant factors were investigated using distributed lag nonlinear models (DLNM) and generalized additive models (GAM).

## 2. Materials and methods

### 2.1. Data collection

**Ethics statement.**    The present study was approved by Shenyang Center for Disease Control and Prevention (CDC). All the HFRS data were anonymously analyzed for the consideration of confidentiality.

**Setting.** Shenyang is the capital city of Liaoning Province. It is a district-level city in China, covering both urban and rural locations. Shenyang is located in latitude 41°11′–43° 02′N and longitude 122°25′–123°48′E, measures 12,860 Sq km and composed of 13 districts and 214 towns [25]. In 2019, Shenyang City's average population was 7,511,923. Shenyang city belongs to the temperate semi-humid continental climate zone. The geographical situation of Shenyang is indicated in S1 Fig.

## The HFRS dataset

We obtained surveillance data on cases of HFRS in Shenyang between 2005 and 2019 from CDC of Shenyang. All patients were diagnosed according to the Criteria and Management Principles of Renal Syndrome Hemorrhagic Fever issued by the Ministry of Health of the People's Republic of China (Ministry of Health 1998). The number of HFRS morbidity was diagnosed according to *Diagnostic Criteria and Principles of Management of Epidemic Hemorrhagic Fever* (GB 15996–1995). For an HFRS case to be confirmed, the affected person must have either traveled to an endemic area or had contact with rodent feces, saliva, or urine within the two months preceding the onset of their illness. They must have experienced an acute illness characterized by abrupt onset of at least two of the following clinical features: fever, chills, hemorrhage, headache, back pain, abdominal pain, acute renal dysfunction, and hypotension. Furthermore, they must have received at least one laboratory confirmation test, such as a positive result for hantavirus-specific immunoglobulin M, a four-fold increase in titers of hantavirus-specific immunoglobulin G between acute and convalescent serum samples, positive hantavirus-specific ribonucleic acid results by reverse transcription polymerase chain reaction in clinical specimens, or hantavirus isolated from clinical specimens, to meet the diagnostic criteria for HFRS [21,26].

## Pollution and meteorological factors data

We obtained daily weather data for 2005 to 2019 from the China Meteorological Data Sharing Service. The weather dataset is available from China Meteorological Data Sharing Service System (http://data.cma.cn). Meteorological factors include air pressure, sunshine, air temperature, air humidity, wind speed and rainfall. In 2013, the national air pollution population health impact monitoring project was officially launched, and Liaoning Province, one of the 16 provinces with heavy air pollution levels, also participated in the air pollution population health impact monitoring project, and Shenyang became the first monitoring city in Liaoning Province. Pollutants information for 2014 to 2019 were originally from 11 state-controlled environmental air quality automatic monitoring stations through the website of Shenyang Bureau of Ecology and Environment due to ensure the accuracy [27]. Pollutants include $PM_{2.5}$, $PM_{10}$, $SO_2$, $NO_2$, CO and $O_3$.

## 2.2. SARIMA and Holt-Winters model construction

Based on the quarterly and monthly data series, additive and multiplicative models were adopted to build factor decomposition models respectively, followed by the application of simple central moving average method to decompose the following four factor maps respectively: (1) long-term trend. (2) cyclical fluctuations. (3) seasonal variations. (4) random fluctuations. Auto regressive integrated moving average model [10] is a time series forecasting method proposed by Geogre Box and Gwilym Jenkins. The ARIMA model is a classical time-series analysis method and is extensively used. The SARIMA model is developed on the foundation of the ARIMA model. The SARIMA model is based on a further development of the ARIMA model, which is particularly suitable for cases where both trend and seasonality are present in the

series. The SARIMA model is abbreviated as SARIMA (P, D, Q) $_S$, where p and q are the orders of autoregressive and moving average, P and Q are the orders of seasonal autoregressive and moving average, d is the number of variances, D is the number of seasonal variances, and S is the seasonal period and cycle length [9]. The construction of the SARIMA model is shown as following equation:

$$\Phi(L)A_P(L^s)(\nabla^d \nabla_s^D x_i) = \Theta_q(L)B_q(L^s)\varepsilon_t$$

$$E(\varepsilon_t) = 0,\ Var(\varepsilon_t) = \sigma_s^2, E(\varepsilon_t \mid \varepsilon_s) = 0, s \neq t$$

$$E(x_s\varepsilon_t) = 0, s < t$$

where L is the delay operator, $A_P(L^s)$ is the p-order autoregressive operator, $A_q(L^s)$ is the q-order seasonal moving average operator, $\nabla^d = (1-L)^d$ is the difference operation, and $\nabla D\ s = (1-L^s)^d$ is the seasonal difference operation. The order of approximation of the model is determined based on the autocorrelation function. The QAIC information criterion is then used to determine the best combination of parameters for the model, and the model satisfies the residual white noise test [28].

The Holt-Winters model is a forecasting technique proposed by Holt and Winters in 1960 that is based on speculative smoothing. Unlike ARIMA, Holt's linear equation has a built-in equation for seasonal factors that directly captures seasonality [29].

Three smoothing equations are used to calculate and evaluate deseasonalized series, trends and seasonality variables. The Holt–Winters' additive method can be written as follows:

$$L_t = \alpha(y_t - S_{t-s}) + (1 - \alpha)(L_{t-1} + b_{t-1})$$

$$S_t = \delta(y_t - L_t) + (1 - \delta)S_{t-1}b_t = \gamma(L_t - L_{t-1}) + (1 - \gamma)b_{t-1}$$

$$S_t = \delta(y_t - L_t) + (1 - \delta)S_{t-1}$$

where $t = 1,. . ., n$, $S$ represents the length of seasonality (months), $L_t$ represents the level of the series, and $b_t$ denotes the trend and seasonal components. The constants used in this model are $\alpha$ (horizontal smoothing constant), $\gamma$ (trend smoothing constant) and $\delta$ (seasonal smoothing constant) [30].

## 2.3. BRT model construction

BRT methods have been successfully applied to research fields such as disease modeling [31]. The BRT method produces a series of trees, each of which grows on the remnant of the previous tree. Recent studies have shown that the BRT model can explain the interactions between exposures in observational studies [32]. In the BRT model, f($x$) is an evaluation of the response $y$ based on a vector predictor of $x$, which in turn is integrated as an additive form of b($x$; $\gamma_m$), as follows:

$$f(x) = \sum_m f_m(x) = \sum_m \beta_m b(x; \gamma_m)$$

where $\beta_m$ is the expansion factor and $b(x; \gamma_m)$ represent the individual trees with parameters $y$ and variables $x$. The coefficient $\beta_m$ reflects the weights allocated to the nodes of each tree and identifies the type of combination predicted for each tree. In this approach, the three regularization parameters, number of trees, learning rate (lr) and tree complexity (tc), should be optimized. The complexity (tc) should be optimized [33]. To this purpose, in this study, various nt,

tc (1–10), and lr (0.001, 0.05, and 0.01) are allocated to the training of the BRT model in order to maximize the model performance [34].

## 2.4. DLNM model and GAM construction

DLNM has been extensively used to assess the exposure-lag-response relationship between environmental factors and human diseases, such as congenital heart disease, HFRS, non-accidental deaths and so on [35,36]. The model can be written as follows:

$$log[E(Y_t)] = \alpha + NS(M, df, lag, df) + \Sigma NS(X_i) + NS(\text{Time}, df) + \beta DOW_t + \gamma \text{Holiday}_t$$

To analyze the lag and extreme effects of environmental factors, Humidity, wind speed and $SO_2$ were taken and applied to the cross-basis functions of DLNM. Here, $Y_t$ was the number of daily counts of HFRS cases in daily t; $\alpha$ was the intercept of the whole model; $NS$ is a natural cubic spline that acts as a smooth function of the model; $M$ represents the estimated environmental variable related to HFRS; $X_t$ is the other environmental variables in the pathogenesis of HFRS that requires nonlinear confounding effect adjustment; $NS$ was used to adjust for daily confounding in the model; $DOW$ is a categorical variable for day of week; Holiday is a binary variable used to control the effect of Chinese public holidays, $\beta$ and $\gamma$ are the regression coefficients; The optimal degrees of freedom ($df$) for the spline function were estimated by Akaike information criterion for quasi-Poisson (Q-AIC) and minimum partial regression coefficient ($PACF_{min}$) criteria; NS of 4,6 and 8 $df$ were used for wind speed, $SO_2$ and relative humidity respectively, and the lag space was set to 3 $df$. NS with 5$df$/year was applied to time variable. In addition, as the incubation period for human hantavirus infection is typically 7–14 days, our model applied the Q-AIC guidelines using a delay of up to 16 days.

In our study, the median environmental variable has been used as a reference value to compute the relative risk. We then compared the 25th and 75th percentile of each environmental variable ("low" and "high") to its median value, in order to explore the stratified effect of modification and qualitatively study the association among environmental factors and HFRS cases. The impact of environmental factors was analyzed by stratifying by gender, age group and number of diagnostic delayed days in order to identify susceptible populations and their corresponding sensitivities.

Subsequently, the GAM method was used to explore the interaction of meteorological and pollutant factors on the prevalence of HFRS. The model can be written as follows:

$$log[E(Yt)] = \alpha 2 + s_1(X_1, X_2) + s_2(X_3) + s_3(\text{day})$$

$\alpha 2$ is the intercept; $X_1$ indicates one of the environmental factors (humidity, wind speed and $SO_2$) whereas $X_2$ and $X_3$ denote the other two; $s$ () indicates penalized spline function. $s_1(X_1, X_2)$ is a spline function of the interaction between the parameters $X_1$ and $X_2$. ($X_1, X_2, X_3$ are all 16 lagged variables.)

## 2.5. Statistical analysis

As there were missing values in the incidence data of HFRS in Shenyang, we performed linear interpolation to compensate as soon as possible in order to better apply Box-Jenkins and exponential smoothing methods for incidence prediction. A total of 10 sets of data were missing, located in the years 2011–2019, mainly in 2016 and 2019. Of these, only a single month was missing in 2011, 2012, 2014, 2015, and 2017, while two months were missing in 2016 (September-October) and three months were missing in 2019 (February, July, and October). For the influence of meteorological and pollutant factors on the number of HFRS cases, Spearman correlation analysis was used for feature selection, followed by BRT to fit the selected features to

the variables and interaction tests. We developed a DLNM with a maximum lag of 16 days to evaluate the lagged, stratification and extreme effects of pollution and meteorological factors on the cases of HFRS. A GAM then was established to explore the interaction of $SO_2$ and two other meteorological factors on HFRS cases.

All analyses in our study were performed in R software (version 4.1.3).

## 3. Results

### 3.1. Descriptive characteristics of HFRS cases and environmental factors

A total of 1,880 cases of HFRS were reported in Shenyang from 2005–2019, of which the incidence of HFRS was predominantly in young adults aged 20–50, accounting for 71.81% of all cases. Men are more susceptible than women at a ratio of 3.67:1 (1477:403). For 2005–2019, the incidence of HFRS in Shenyang was 25.03 per 100,000, and the case fatality rate was 0.691% (Table 1).

The onset of HFRS showed significant differences in seasonality, age, and delayed days in diagnosis of onset ($p<0.05$) (Table 2). From Table 2 meaningful subgroups differed in seasonal distribution, showing that different age groups were mainly concentrated in summer and winter, while groups with different days of delayed onset were mainly concentrated in spring and winter. Summary statistics of all HFRS cases and environmental variables in Shenyang are shown in S1 Table. Fig 1 shows the time series distribution of daily cases of HFRS and environmental factors from 2005–2019. There are distinct seasonal variations in both HFRS and environmental conditions.

### 3.2. Time-series analysis of HFRS% of Holt-Winters and SARIMA model in monthly and seasonal prediction

From the factor decomposition diagrams in S2 Fig, the annual incidence rate of HFRS is trending down, and after removing the trend effect from the original series, the difference in the

**Table 1. Distribution of the hemorrhagic fever with renal syndrome (HFRS) cases by age and season group in Shenyang, 2005–2019.**

| Characteristic | | 0–20 | 20–50 | >50 | Total | Population | Incidence (per $10^5$) | No of Deaths | Mortality (per $10^5$) | Case fatality (%) |
|---|---|---|---|---|---|---|---|---|---|---|
| | | No of HFRS cases (%) | | | | | | | | |
| Year | 2005 | 31(6.33%) | 368(75.10%) | 91(18.57%) | 490 | 6962186 | 7.04 | 4 | 0.057 | 0.816 |
| | 2006 | 18(5.44%) | 250(75.53%) | 63(19.03%) | 331 | 7010640 | 4.72 | 3 | 0.043 | 0.906 |
| | 2007 | 11(5.91%) | 139(74.73%) | 36(19.35%) | 186 | 7066666 | 2.63 | 2 | 0.028 | 1.075 |
| | 2008 | 5(2.78%) | 139(77.22%) | 36(20%) | 180 | 7116384 | 2.53 | 0 | 0.000 | 0.000 |
| | 2009 | 6(4.65%) | 88(68.22%) | 35(27.13%) | 129 | 7150272 | 1.80 | 1 | 0.014 | 0.775 |
| | 2010 | 6(6.38%) | 60(63.83%) | 28(29.79%) | 94 | 7180769 | 1.31 | 0 | 0.000 | 0.000 |
| | 2011 | 5(5.81%) | 59(68.60%) | 22(25.58%) | 86 | 7211479 | 1.19 | 0 | 0.000 | 0.000 |
| | 2012 | 3(3.70%) | 59(72.84%) | 19(23.46%) | 81 | 7237420 | 1.12 | 0 | 0.000 | 0.000 |
| | 2013 | 1(1.37%) | 49(67.12%) | 23(31.51%) | 73 | 7259528 | 1.01 | 0 | 0.000 | 0.000 |
| | 2014 | 0(0%) | 28(54.90%) | 23(45.10%) | 51 | 7289761 | 0.70 | 0 | 0.000 | 0.000 |
| | 2015 | 1(2%) | 35(70%) | 14(28%) | 50 | 7306224 | 0.68 | 0 | 0.000 | 0.000 |
| | 2016 | 2(6.06%) | 25(75.76%) | 6(18.18%) | 33 | 7324009 | 0.45 | 1 | 0.014 | 3.030 |
| | 2017 | 1(2.22%) | 23(51.11%) | 21(46.67%) | 45 | 7356745 | 0.61 | 0 | 0.000 | 0.000 |
| | 2018 | 3(8.33%) | 21(58.33%) | 12(33.33%) | 36 | 7414719 | 0.49 | 2 | 0.027 | 5.556 |
| | 2019 | 1(6.67%) | 7(46.67%) | 7(46.67%) | 15 | 7511923 | 0.20 | 0 | 0.000 | 0.000 |
| Seasons | Spring(Mar-May) | 14(4.53%) | 209(67.64%) | 86(27.83%) | 309 | - | - | 4 | - | 1.294 |
| | Summer(Jun-Aug) | 33(4.51%) | 543(74.18%) | 156(21.31%) | 732 | - | - | 3 | - | 0.410 |
| | Autumn(Sep-Nov) | 24(7.08%) | 223(65.78%) | 92(27.14%) | 339 | - | - | 2 | - | 0.590 |
| | Winter(Dec-Feb) | 23(4.60%) | 375(75%) | 102(20.40%) | 500 | - | - | 4 | - | 0.800 |
| Total | | 94(5%) | 1350(71.81%) | 436(23.19%) | 1880 | 7511923 | 25.03 | 13 | 0.173 | 0.691 |

**Table 2. Stratified characteristics of the hemorrhagic fever with renal syndrome (HFRS) seasonal cases of Shenyang.**

| Characteristic | | Spring | Summer | Autumn | Winter | Total | *P*-value |
|---|---|---|---|---|---|---|---|
| | | No of HFRS cases (%) | | | | | |
| Age | 0–20 | 14(4.53%) | 33(4.51%) | 24(7.08%) | 23(4.60%) | 94 | <0.05 |
| | 20–50 | 209(67.64%) | 543(74.18%) | 223(65.78%) | 375(75%) | 1350 | |
| | >50 | 86(27.83%) | 156(21.31%) | 92(27.14%) | 102(20.40%) | 436 | |
| The interval of days | 0–4 | 428(38.35%) | 219(19.62%) | 169(15.14%) | 300(26.88%) | 1116 | <0.05 |
| | 5–9 | 248(42.18%) | 96(16.33%) | 94(15.99%) | 150(25.51%) | 588 | |
| | 10- | 56(31.82%) | 24(13.64%) | 46(26.14%) | 50(28.41%) | 176 | |
| Occupation | Farmer | 210(33.49%) | 135(21.53%) | 109(17.38%) | 173(27.59%) | 627 | >0.05 |
| | Home | 219(35.44%) | 116(18.77%) | 119(19.26%) | 164(26.54%) | 618 | |
| | Other | 303(47.72%) | 88(13.86%) | 81(12.76%) | 163(25.67%) | 635 | |

average seasonal index among different quarters is the difference caused by the seasonal effect. HFRS in Shenyang peaked twice a year in the first and fourth quarter. The monthly HFRS has the characteristics of bimodal distribution, with the first peak in March-May, the second peak in November-December, and the seasonal inelasticity rising from April-August. The incidence of HFRS in Shenyang is characterized by a bimodal monthly distribution, with the first peak in March-May, the second peak in November-December, and an exponential decline in April-August, and a seasonal inverse rise starting in September.

The model fixed-order plots in S3 Fig allow the SARIMA seasonal and monthly models to be parameterized for the ACF, PACF plots, combined with the "auto.arima()" function to correct for the AR and MA parameters (parameter estimates < 2 times the sample standard deviation). The model parameters and tests are shown in Table 3, it gives the forecasting accuracy of two models for the HFRS series. The SARIMA model has lower values for RMSE, MAE and MAPE, which means the SARIMA is more accurate.

After building a suitable model based on the model parameters, the incidence of HFRS from 2005–2018 was used as the training set and 2019 was used as the validation set to validate the model established by the training set. The Holt-Winters and SARIMA models were applied to predict the incidence of HFRS, respectively, and the prediction effects were plotted as obtained in Fig 2 and S2 Table. From them the Holt-Winters model predicts trends closer to the actual values than the SARIMA model. the 95% confidence interval for the SARIMA model is narrower than the Holt-Winters model and its interval contains all the actual values. S4 and S5 Figs show the tests of goodness of fitness and significance for the series of HFRS incidence from the two methods.

### 3.3. Feature selection and fitting of environmental factors for HFRS

Spearman correlation analysis showed that HFRS was significantly correlated with humidity (r = -0.10, p<0.01), wind speed (r = 0.07, p<0.05) and $SO_2$ (r = 0.09, p<0.01) (S3 Table). Furthermore, to fit the BRT model, we set the model parameters: tree complexing was 5, learning rate was 0.005, and bag. fraction was 0.5. According to S6 Fig, the degree of fit of each of the three environmental factors fitting functions was like the Spearman correlation results, and the trend with the number of HFRS incidence was significant.

Depending on the distribution of observations in the environment space, the fitting function can give a distribution of fitted values relating to each predictor. The values at the top of each graph indicate the weighted average of the fitted values associated with each non-factor predictor. According to the interaction fitting function in Fig 3A and 3B, environmental factors fit better at moderate levels of interaction.

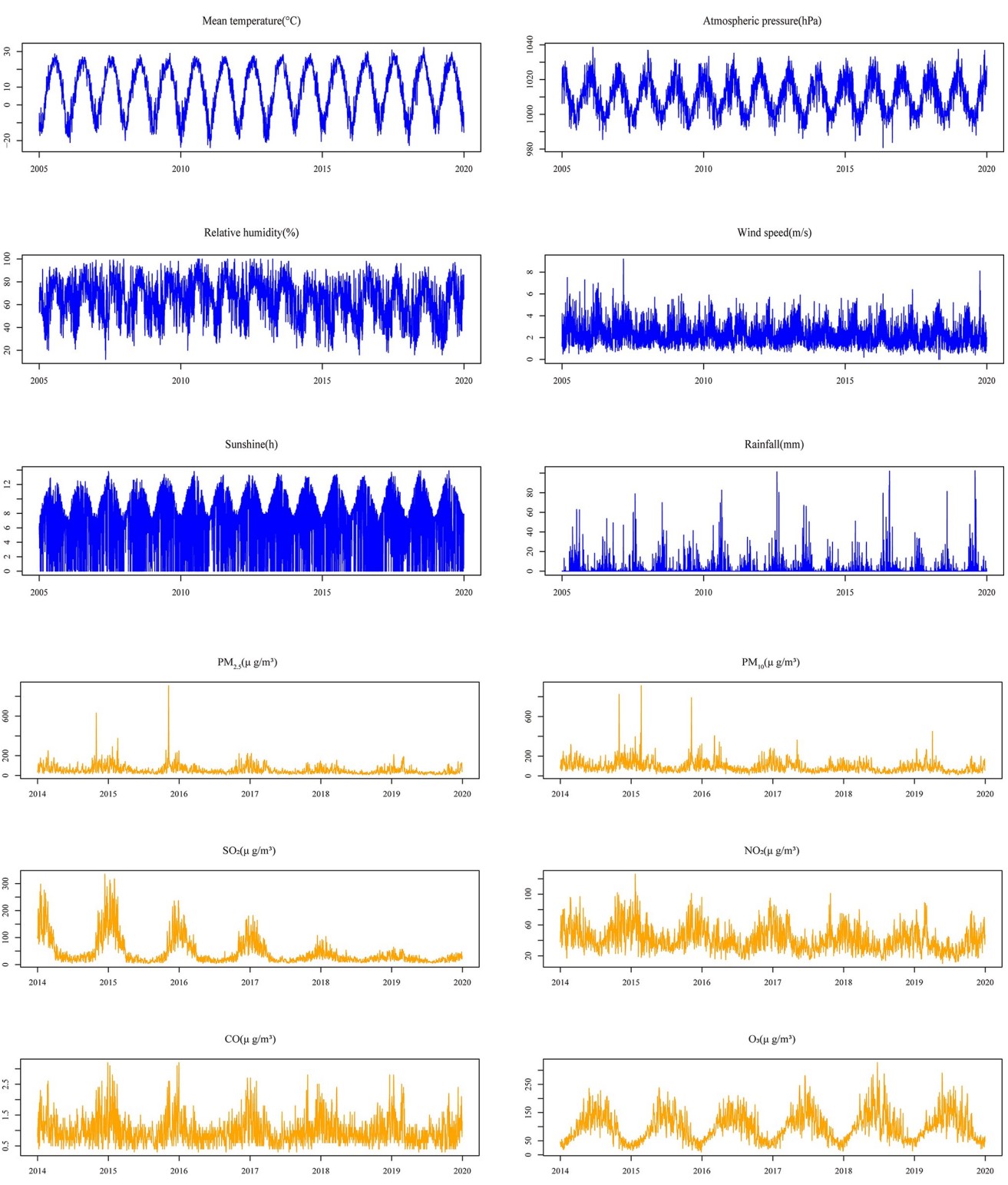

**Fig 1. The time series distribution of daily HFRS cases, meteorological and air-pollution factors in Shenyang from 2005–2019.**

**Table 3. Performance measures of time series techniques for the hemorrhagic fever with renal syndrome (HFRS) incidence in Shenyang.**

| Model | Best parameters | Method | Box-Ljung test | | | RMSE | MAE | MAPE |
|---|---|---|---|---|---|---|---|---|
| | | | X-squared | Df | p-value | | | |
| Holt-Winters seasonal model | $\alpha = 0.27$, $\beta = 0.38$, $\gamma = 1$ | Additive | 0.087 | 2 | 0.957 | 0.198 | 0.13 | 40.215 |
| Holt-Winters monthly model | $\alpha = 0.24$, $\beta = 0.05$, $\gamma = 0.75$ | Additive | 7.386 | 6 | 0.287 | 0.078 | 0.054 | 71.535 |
| SARIMA seasonal model | $((1,2),1,0)\ (0,1,0)$ [4] | Additive | $6.59 \times 10^{-6}$ | 1 | 0.998 | 0.182 | 0.116 | 38.031 |
| SARIMA monthly model | $(1,1,0)\ (2,1,0)$ [12] | Multiple | 0.381 | 1 | 0.537 | 0.071 | 0.049 | 65.968 |

### 3.4. The lag relationship between environmental variables and the incidence of HFRS

S7 Fig shows that the non-linear exposure–lag–response association among daily humidity, wind speed, $SO_2$ and HFRS incidence cases, which indicated that these factors are relatively high risk above their median levels at different lag days. Different lag times correspond to different effects, specifically the effect of low wind speed occurs rapidly but lasts for a short time, the effect of high wind speed has a longer lag time but has a greater impact, while high humidity can have a transient effect on HFRS and high levels of $SO_2$ can have a transient or continuous effect on HFRS, with the effect initially concentrated in the lag time of 0–5 days and after 15 days.

Estimates of the impact of meteorological and pollutant factors on HFRS cases show varied lagging features. S8 Fig shows the overall effect of environmental variables for total, gender, age and delayed-days HFRS cases within 16 days. Overall, these meteorological and pollutant variables were significantly relevant to HFRS cases. We found that RRs increased with the

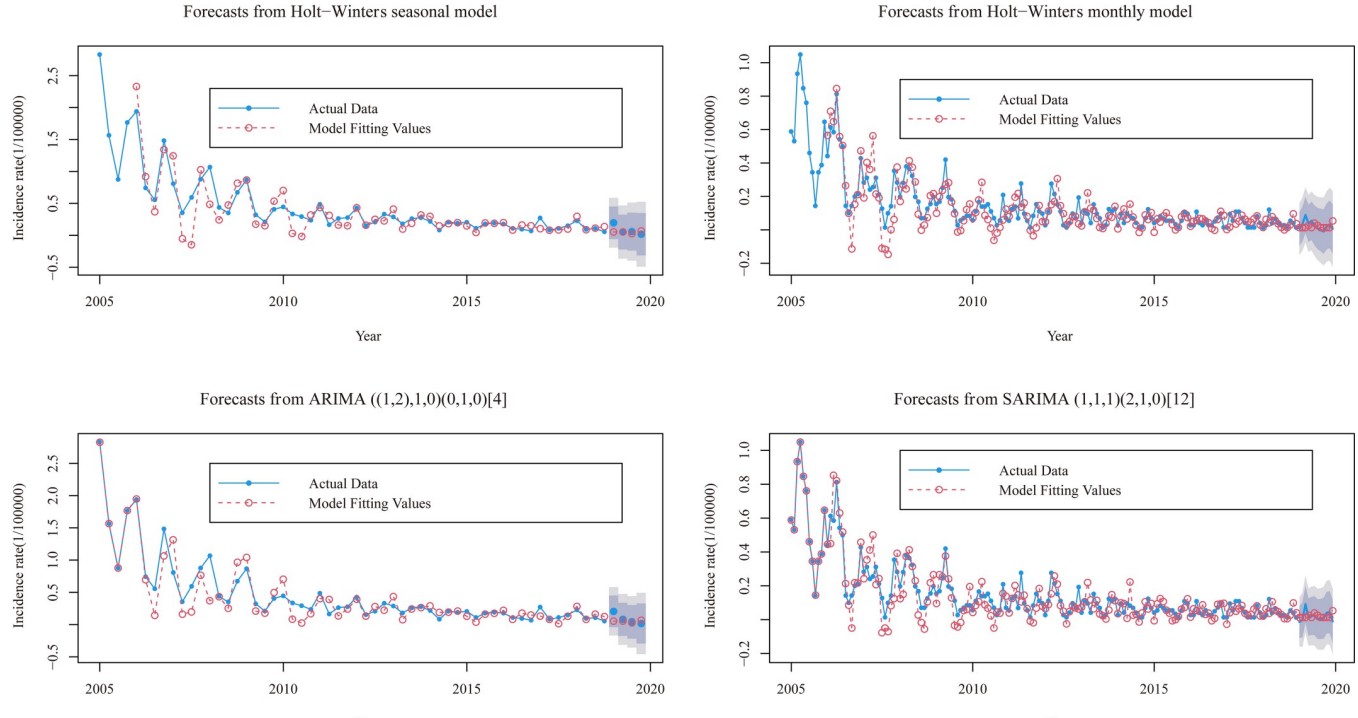

**Fig 2. Resulting comparisons of the HFRS seasonal and monthly incidences using the preferred two models.** The deep shaded regions indicate 80% confidence intervals, the light shaded regions indicate 95% confidence intervals.

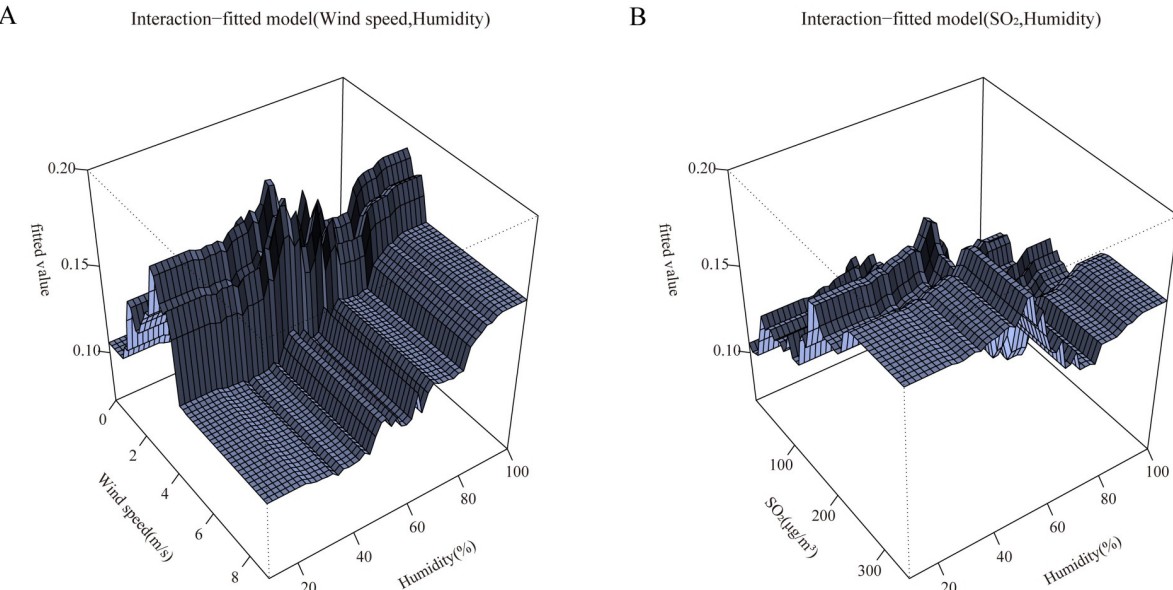

**Fig 3. The fitting interactions of the association among humidity, windspeed, SO$_2$ and HFRS cases in Shenyang, 2014–2019 based on the boosted regression tree models.**

improvement of humidity, wind speed, and SO$_2$, suggesting that higher humidity, wind speed, and SO$_2$ increased the risk of HFRS. Yet humidity and SO$_2$ separately reached the peak at 98% and 229.1μg/m$^3$, then began to decrease or stabilize. Wind speed peaked at 3m/s and decreased. In general, similar trends in exposure-response relationships between environmental variables and cases of HFRS disease by gender, age and delayed-days group compared to total cases are shown in S8 Fig. The minimum risk of incidence (RR$_{min}$) values for environmental factors such as humidity, wind speed and SO$_2$ were 16%, 8.1m/s and 223μg/m$^3$ respectively.

Generally, analogous trends in exposure-response and lag-response relationships among environmental variables and HFRS cases across gender, age, and delay days of groups compared to total cases are shown in Figs 4, 5 and S8.

### 3.5. Exposure-response relationships for environmental factors with different lag times

As shown in Fig 4, the effects of humidity and SO$_2$ on HFRS differed across lag times and stratification factors when the study lag time points were 0 and 16 days. The RR of the effect of humidity on HFRS cases tended to increase at lag 0 and 16 days, with humidity RR values reaching a maximum at 20–40% and above 90% at lag 0, while lag 16 days only showed a maximum RR value at high humidity. The effect values of wind speed and SO$_2$ on HFRS cases at lags 0 and 16 days showed a trend of increasing and then decreasing RR, with RR values in the range of 2-4m/s and 200–250μg/m$^3$, respectively. Within the different grouping intervals, the trend of RR effect values for humidity within Delay 0–4 days was slightly different from the overall, and the RR effect values for wind speed within Female, Delay 0–4, Delay 5–9 days, and SO$_2$ within Delay 0–4 days were significantly different.

### 3.6. Effects of extreme environmental variables on HFRS cases

To determine the effect of extreme environmental factors on the HFRS, the estimated effects were examined by comparing the 25th or 75th percentile of relative humidity, wind speed and

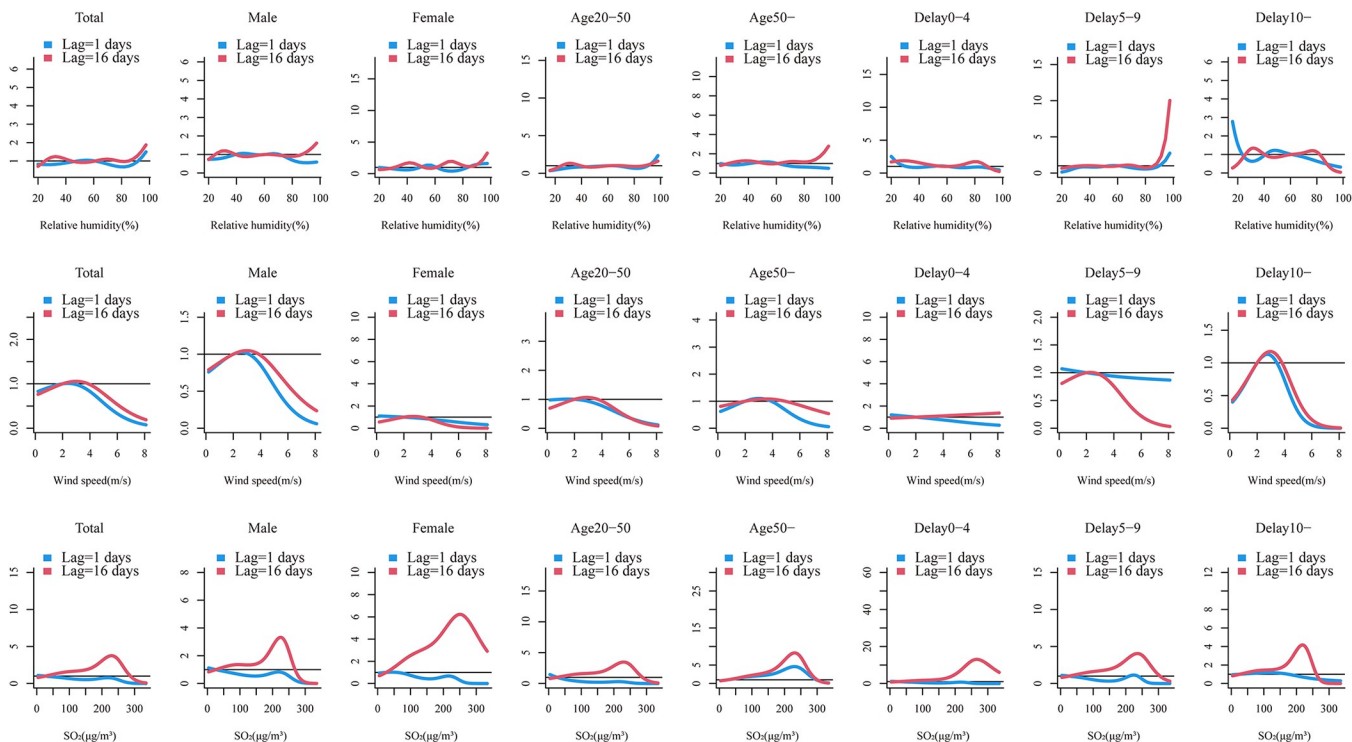

**Fig 4. Summary of slices lag1-16 days exposure-response relationship between meteorological factors, air-pollution and HFRS cases for total, gender (male, female), age (20-50years, and 50-years) and delay groups (0-4days, 5-9days and 10-days) in Shenyang.**

$SO_2$ with their median values. Table 4 shows the cumulative impact of the lag factor extremes on the HFRS at 16 lag days. We found that extreme high levels of $SO_2$ were positively linked to the onset of HFRS, while extreme low levels of $SO_2$, with no wind effect, had a protective effect, and the RR values of cumulative effects were 2.583 (1.145,5.827) for high $SO_2$ effect and 0.577 (0.370, 0.898) for cold $SO_2$ effect. At 16 lag days, significant cumulative effects of low windy conditions were observed in males (RR value: 0.490 (0.241,0.997)), in the over 50 years age group (RR value: 0.335 (0.113,0.992)) and in delayed onset for over 10 days (RR value: 0.324 (0.106,0.983)). In turn, women at extreme $SO_2$ levels and patients with a delayed onset of 5–9 days are susceptible, with their RR values: 8.122 (1.009,65.403) for high $SO_2$ and 0.285 (0.090,0.898) for low $SO_2$, 4.491(1.246,16.193) for high $SO_2$ and 0.427(0.213, 0.858) for low $SO_2$, respectively.

The distributed lagged effects of extreme environmental factors at various lag days for all groups were showed in Fig 5. We found that the dry effect indicated a maximum RR value on the current day, peaking at 4 lag days and then showed a U-shaped curve along the lag days, and the RR value subsequently decreased for the next days and then turned to rise along the lag days, whereas wet effect showed the opposite trend. The curve of dry and wet effect was roughly similar among different stratified groups. On the low windy effect, the 2 lag days is a peak followed by a decline, while the opposite is true for the high windy effect. The overall patients reached their highest effect at extreme high or low levels of $SO_2$, usually at a lag of 1 day, followed by a gradual downward trend. female, aged >1 years and delayed 5–9 days remain the most sensitive people.

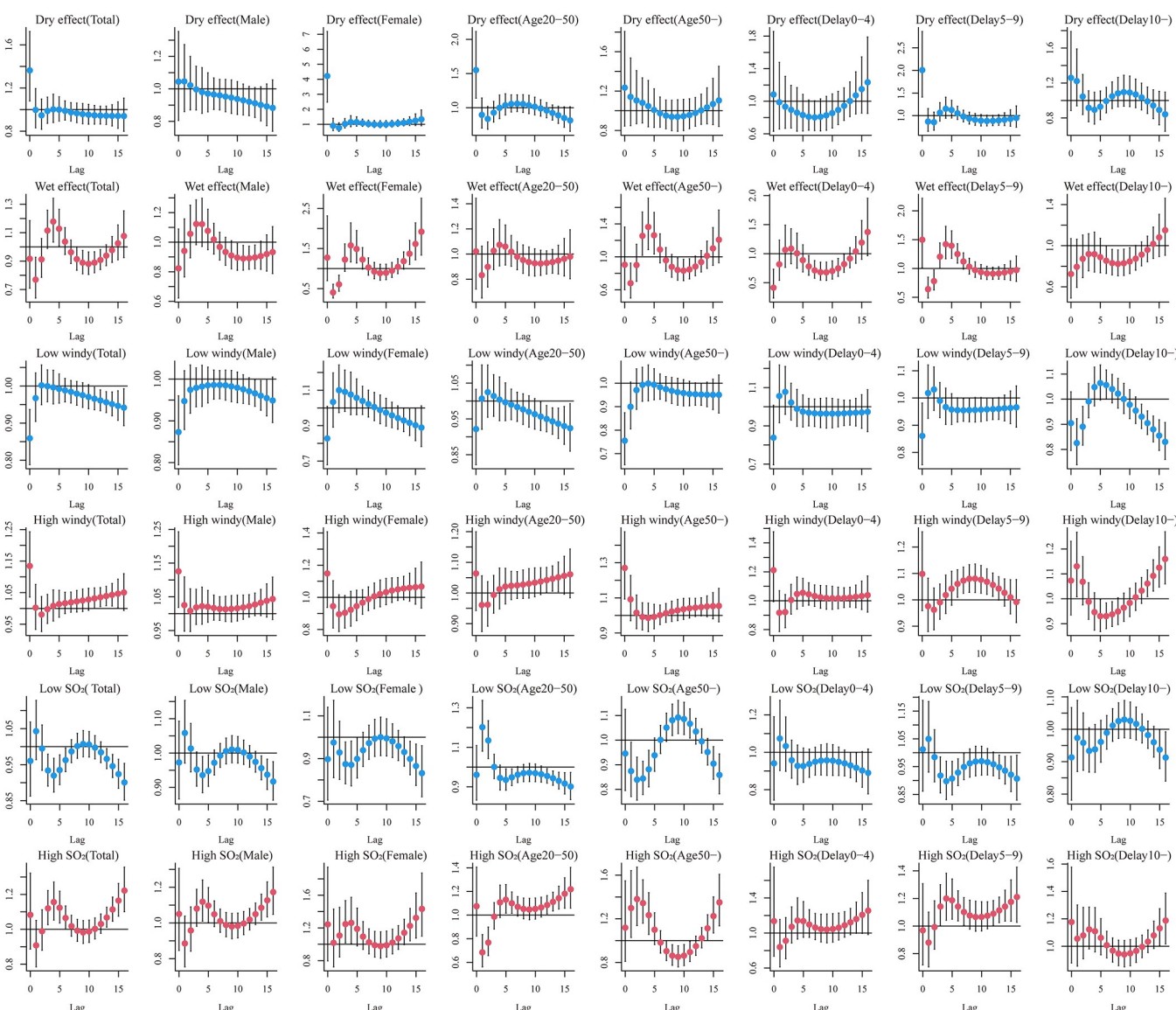

**Fig 5. Summary of estimated extreme effects at the 25th and the 75th percentile of relative humidity, wind speed and SO₂ on HFRS cases for total, gender (male, female), age (20-50years, and 50-years) and delay groups (0-4days, 5-9days and 10-days) at different lag days.** The median value of each meteorological factor (relative humidity: 61%, wind speed: 2 m/s, SO₂: 26g/m³) is used as a reference level.

## 3.7. Environmental interaction during humidity, wind speed, SO₂ and HFRS cases

GAMs were built to show the interaction effect among humidity, wind speed and SO₂ on HFRS incidence (Fig 6). The program on the top side of Fig 6A and 6B shows the interaction effect of wind speed and humidity on HFRS. The HFRS infection risk increased as daily wind speed and humidity decreased. The plot to the bottom of Fig 6C and 6D indicates the interaction effect of SO₂ and humidity, HFRS tends to occur in higher SO₂ and lower humidity environmental conditions.

## 4. Discussion

From the time series and seasonal decomposition of the incidence of HFRS in Shenyang, combined with studies in various regions of China [37], it can be seen that there is a clear seasonal

**Table 4. The cumulative effects of extreme (25ᵗʰ and 75ᵗʰ percentile vs. median level) meteorological and air-pollution factors on HFRS cases of children by sex, age and delay.**

| Series | Variables | Cumulative effects(95%CI) | | | | | |
|---|---|---|---|---|---|---|---|
| | | Dry effect | Wet effect | Low windy effect | High windy effect | Low SO₂ effect | High SO₂ effect |
| | Total cases | 0.761(0.158,3.673) | 0.557(0.129,2.405) | **0.487(0.260,0.912)** | 1.617(0.982,2.663) | **0.577(0.370,0.898)** | **2.583(1.145,5.827)** |
| Sex | Male | 0.461(0.078,2.737) | 0.462(0.088,2.428) | **0.490(0.241,0.997)** | 1.593(0.906,2.801) | 0.692(0.425,1.126) | 1.832(0.750,4.474) |
| | Female | 9.817(0.203,473.684) | 4.091(0.131,127.899) | 0.683(0.153,3.039) | 1.075(0.347,3.334) | **0.285(0.090,0.898)** | **8.122(1.009,65.403)** |
| Age | 20–50 years | 0.745(0.094,5.870) | 0.535(0.083,3.451) | 0.523(0.225,1.213) | 1.544(0.787,3.032) | 0.645(0.363,1.146) | 2.031(0.706,5.845) |
| | 50- years | 1.574(0.097,25.407) | 0.723(0.053,9.803) | **0.335(0.113,0.992)** | 2.007(0.855,4.711) | 0.530(0.240,1.167) | 3.287(0.767,14.079) |
| Delay | 0–4 days | 0.303(0.005,19.164) | 0.061(0.002,2.207) | 0.589(0.129,2.677) | 1.515(0.460,4.993) | 0.408(0.126,1.319) | 3.652(0.438,30.408) |
| | 5–9 days | 0.897(0.088,9.191) | 1.540(0.175,13.525) | 0.451(0.180,1.133) | 1.874(0.884,3.971) | **0.427(0.213,0.858)** | **4.491(1.246,16.193)** |
| | 10- days | 1.285(0.094,17.643) | 0.149(0.012,1.899) | **0.324(0.106,0.983)** | 1.392(0.596,3.254) | 0.638(0.314,1.297) | 2.104(0.570,7.762) |

Bold font indicates statistical significance at the 0.05 level.

trend in the incidence of HFRS in China, with a decreasing trend year upon year. The peak in case reporting differs between single and double peaks across regions, with the main peak occurring between March and May each year. This study also shows a second peak in November to December, which may occur for the following reasons: (1) The peak of population movement is about March each year, when the mobility of urban life becomes more complex and the fast pace of life and consumption leads to a gradual decline in the demand for health. (2) After November, the cooler temperatures in the city lead to larger crowds and a greater temperature difference between indoors and outdoors, allowing host animals to enter human life more closely and the disease to be more contagious.

Currently, many scholars have conducted research on predictive models for the onset of infectious diseases. The SARIMA model had been used to predict the incidence of HFRS [13,38]. It shows that SARIMA has the characteristics of being unconstrained by data type and high applicability, integrating factors such as trend, periodicity and random error, so that it can be used in prediction studies of infectious diseases with periodic morbidity characteristics. Pritthijit Nathet al. applied both the SARIMA model and the Holt-winters seasonal model for the prediction of airborne particulate matter in eastern India, the Holt—winters model was considered to be simple in principle and had a high predictive accuracy for diseases with a cyclical pattern of onset [39]. In this study we discussed the effect of the SARIMA model applied to the HFRS series and compared it with the Holt-Winters model. In terms of model mechanism, the SARIMA model is suitable for predicting series that are smooth and stable over time compared to the Holt—winters model, which is suitable for predicting models with a single trend of change. However, research on time series has limitations, as both methods in this study are extrapolated forecasts based on historical data, usually considering the characteristics of the series itself, and cannot predict sudden changes in the data due to changes in external factors. Moreover, the occurrence and prevalence of infectious diseases are influenced by multiple natural factors, climate and other social factors. Our study thus explored the lagging, interactive and stratified effects of meteorological and pollutant factors on the prevalence of HFRS in Shenyang.

Firstly, we need to select and fit the features using the spearman method combined with a BRT. The results were chosen from three environmental factors, wind speed, humidity and SO₂, in order to achieve a precise study of the influence of environmental factors on HFRS. Subsequently, the DLNM method was applied to examine the exposure-lag-response relationship between the average daily cases of HFRS disease and environmental factors in Shenyang from 2014–2019. The results showed a non-linear lagged relationship among meteorological, pollutant factors and HFRS. Extremely high concentration levels of SO₂ increased the risk of

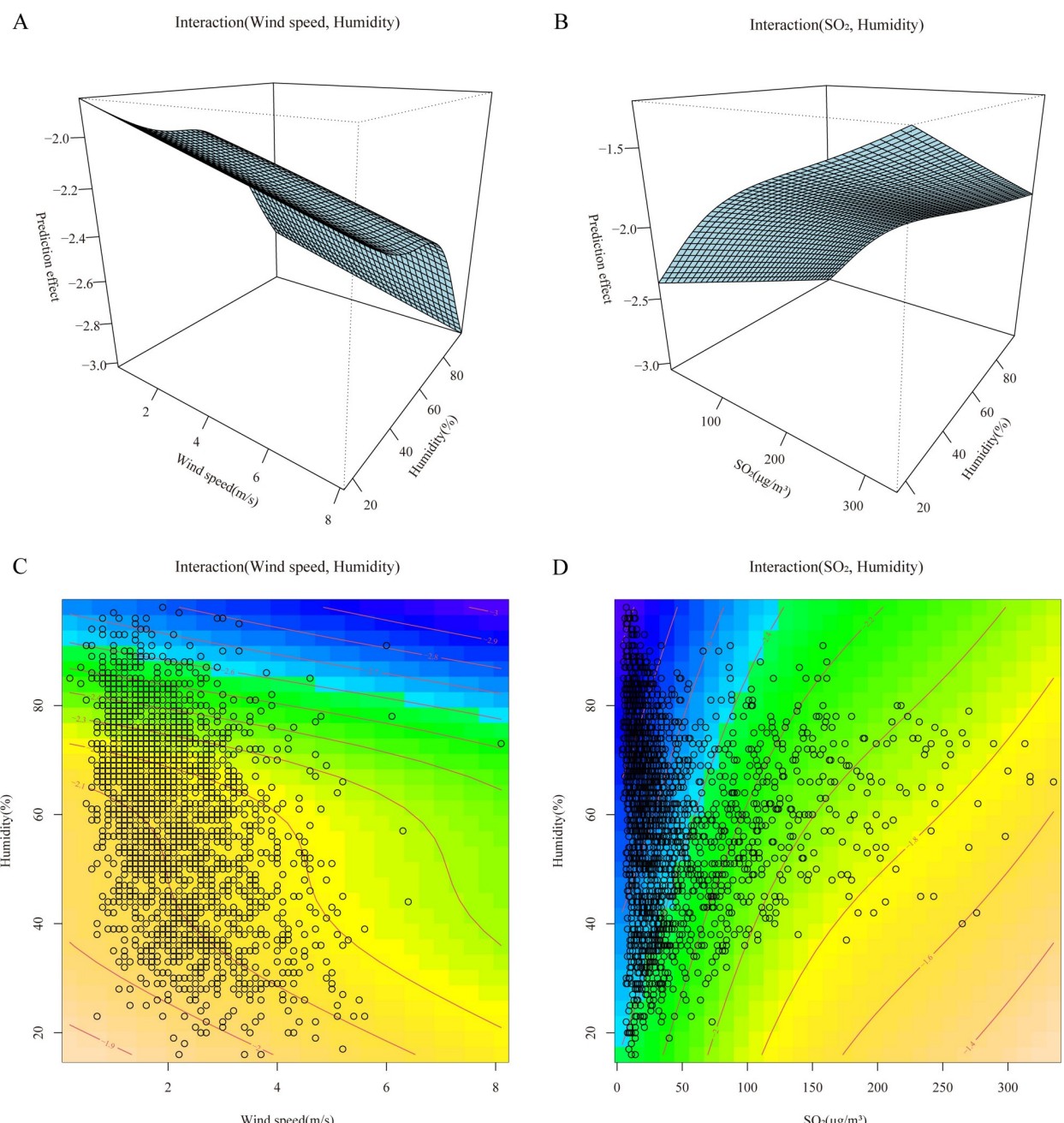

**Fig 6. The effect interactions of the association among humidity, windspeed, SO₂ and HFRS cases in Shenyang, 2014–2019 based on the generalized additive models.**

HFRS, while low windy and low concentration levels of SO$_2$ were protective against HFRS from 0-16d. It was also found that the lagged effects of different climatic and pollutant factors were not identical. The different delay periods reflect the fact that the lagged effect of each environmental variable may be related to the spread of infection influenced by various factors, including the growth of the virus in the external environment, the inclination of people to go outside, and seasonal changes in rodent populations [21,40]. We demonstrated that high concentrations of SO$_2$ significantly influenced the spread of HFRS after 0–5 and 15 lag days.

Extreme low windy were strongly associated with HFRS from lag 0 to a maximum lag 16 days, suggesting that the incidence of HFRS may be lagged by approximately 16 days at low windy. It could be similar to the study by Zhang [41] et al. on HFMD, mainly because low windy may inhibit the spread of hantavirus-containing particles [42]. The effect of humidity on HFRS cases tended to increase at days 0 and 16 when the lag time was 16 days, with the greatest effect at 20–40% at lag 0 days and at over 90% at lag 16 days. Higher humidity levels may indicate that humidity affects the survival of the rodent host, in addition to affecting the infection and stability of the virus in the in vitro environment [21]. Furthermore, in contrast to previous studies, our study included pollutants in the HFRS influencing factors and explored the non-linear lag between $SO_2$ and HFRS. Our study found that elevated $SO_2$ concentrations increased the risk of HFRS infection at levels in the range of 200–250 μg/m$^3$ and were significant in women and in patients with a delayed onset of 5–9 days. There is still controversy about the effect of $SO_2$ on HFRS, which may be related to regional, population differences and the proportion of pollutants in the air. However, regarding the effect of $SO_2$ on other infectious diseases, there were different reports showing a significant protective effect of $SO_2$ against influenza [43](RR = 0.892, 95% CI: 0.840–0.948), which probably due to the higher outdoor pollutants, resulting in a population more dependent on the indoor environment and less exposed to the virus.

Stratified analysis showed that the effects of meteorological and pollutant factors varied by sex, age group, and number of days delayed onset. Men were more sensitive to extreme low windy than women, and women were more sensitive to extreme $SO_2$ concentration levels than men. Similar results have been reported in several studies on other infectious diseases [43]. Patients over 50 years were more significantly affected by extreme low windy and showed a protective effect, as compared to other age groups. But based on previous studies, wind speed effects were generally significantly associated with lower age [44]. This study showed that HFRS mainly affected people under 50 years of age at low windy, which might be attributed to underlying factors such as social factors, population distribution, etc. In terms of delay days, patients with a delay of 5–9 days were more sensitive to extreme $SO_2$ concentrations and patients with a delay of over 10 days were more susceptible to extreme low wind speed. The delay between the onset of HFRS and the time of diagnosis led to a lagging effect of environmental factors reinforced by the length of time the patient spent in the environment after onset. Additionally, we did not take temperature, barometric pressure into account in our study of the correlation between environmental factors and HFRS. Some studies had shown that mean and extreme temperatures were negatively correlated with cases of HFRS [20]. This study did not discuss the relationship with HFRS cases in terms of temperature, barometric pressure, and rainfall factors, showing different findings of HFRS in Shenyang before 2011 and after 2014, suggesting possible spatial and temporal variability.

The results of the interaction analysis showed that higher $SO_2$ and lower humidity environments were the dangerous environmental conditions for the occurrence of HFRS. It was demonstrated that $NO_X$ and $SO_2$ in the air showed strong seasonal variations and that their concentrations were closely related to meteorological factors such as wind speed, temperature and relative humidity. Air pollution may impact the frequency of HFRS cases by modifying viral infectivity and immunity in humans and rodents [45,46].The combined effect of low windy and low humidity also affected the development of HFRS disease. Analysis of the geographical distribution of the country suggested that this result could be attributed to the region's location in a climatic zone [47].

Our research benefits cover: (1) the study period is long, and the study collected case and environmental data over many years. (2) For the time series analysis, we applied two different models for comparison and also split the data into monthly and seasonal data for accurate comparison and forecasting. (3) Our study applies advanced statistical methods, not only

applying spearman to feature selection, but also applying the BRT method to fit the screened variables and their interactions, followed by DLNM and GAM to analyze the lagged, extreme and cumulative effects of environmental factors. Our findings can provide evidence and guidance on the lagged effects and interactions of environmental factors on HFRS. It is worth pointing out that there were some limitations to our study. Firstly, there are cases of HFRS in this study that have been diagnosed both clinically and through the laboratory, which may be subject to diagnostic bias and are under-reported. Moreover, due to the regional limitations of this study, other regions should be referred to with caution in studying the impact and prediction of HFRS disease, considering regional characteristics, and making changes in model selection, parameters and factor selection.

## Supporting information

**S1 Table. Descriptive statistics for daily HFRS cases, meteorological and air-pollution factors in Shenyang, 2005–2019.**
(XLS)

**S2 Table. Comparison of the prediction results of the two models for 2019 in Shenyang.**
(XLS)

**S3 Table. Spearman correlation between daily HFRS cases and the affected variables in Shenyang.**
(XLS)

**S1 Fig. The geographical location of Shenyang City in China.** The map was created by Arc-GIS 10.3 (Environmental Systems Research Institute; Redlands, CA, USA). The base map was acquired from the data center for geographic sciences and natural sources research, CAS (http://www.resdc.cn/data.aspx?DATAID=201).
(TIF)

**S2 Fig. Decomposition of determinants in HFRS cases with seasonal and monthly model.**
(TIF)

**S3 Fig. Autocorrelation function (ACF) and partial ACF charts of seasonal (A, B) and monthly (C, D) HFRS incidence with SARIMA model.**
(TIF)

**S4 Fig. Tests of goodness of fit for the error series of HFRS incidence from the Holt-Winters method.** (a)Autocorrelation function (ACF) plot for the seasonal Holt-Winters residual series; (A)Autocorrelation function (ACF) plot for the monthly Holt-Winters residual series; (b) Partial autocorrelation function (PACF) plot for the seasonal Holt-Winters residual series; (B) Partial autocorrelation function (PACF) plot for the monthly Holt-Winters residual series; (c) Standardized residual seasonal Holt-Winters series; (C) Standardized residual monthly Holt-Winters series. These manifested its adequacy and suitability of this data-driven hybrid model for the data.
(TIF)

**S5 Fig. The significance of SARIMA seasonal(A) and monthly(B) model.**
(TIF)

**S6 Fig. Fitting function variables and distribution of fitted values created based on the boosted regression tree models.**
(TIF)

**S7 Fig. Relative risk of meteorological and air-pollution variables on HFRS incidence over 16 lag days, including relative humidity, wind speed and SO$_2$.**
(TIF)

**S8 Fig. Effect of different meteorological and air-pollution variables on the incidence of HFRS at different days for total, gender (male, female), age (20–50 years, and 50-years) and delay groups (0–4 days, 5–9 days and 10-days) in Shenyang.**
(TIF)

## Author Contributions

**Conceptualization:** Ye Chen, Weiming Hou, Jing Dong.

**Data curation:** Ye Chen, Weiming Hou.

**Formal analysis:** Ye Chen, Weiming Hou.

**Investigation:** Ye Chen.

**Methodology:** Ye Chen, Weiming Hou, Jing Dong.

**Software:** Ye Chen, Weiming Hou.

**Validation:** Ye Chen, Weiming Hou, Jing Dong.

**Visualization:** Ye Chen, Weiming Hou.

**Writing – original draft:** Ye Chen, Weiming Hou, Jing Dong.

**Writing – review & editing:** Ye Chen, Weiming Hou, Jing Dong.

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
