## [Decision Letter · Decision Letter 0]

22 Jan 2023

Dear Dr Dong,

Thank you very much for submitting your manuscript "Prediction, impacts and interaction of meteorological and pollution variables for the development of renal syndrome hemorrhagic fever" for consideration at PLOS Neglected Tropical Diseases. As with all papers reviewed by the journal, your manuscript was reviewed by members of the editorial board and by independent reviewers. The reviewers appreciated the attention to an important topic. Based on the reviews, we are likely to accept this manuscript for publication, providing that you modify the manuscript according to the review recommendations. 

In addition to the reviewers' input, we have attached a PDF with editorial input that must also be addressed in your revisions.

Sincerely,

Christopher M. Barker

Academic Editor

Victoria Brookes

Section Editor

Reviewer's Responses to Questions

**Key Review Criteria Required for Acceptance?**

**Methods**

-Are the objectives of the study clearly articulated with a clear testable hypothesis stated?

-Is the study design appropriate to address the stated objectives?

-Is the population clearly described and appropriate for the hypothesis being tested?

-Is the sample size sufficient to ensure adequate power to address the hypothesis being tested?

-Were correct statistical analysis used to support conclusions?

-Are there concerns about ethical or regulatory requirements being met?

Reviewer #1: Are the objectives of the study clearly articulated with a clear testable hypothesis stated?

-Is the study design appropriate to address the stated objectives? Yes, the study design is appropriate.

-Is the population clearly described and appropriate for the hypothesis being tested? Yes, Sheyang is the population identified.

-Is the sample size sufficient to ensure adequate power to address the hypothesis being tested? The length of the study is sufficient 2005-2019.

-Were correct statistical analysis used to support conclusions? Yes, this was done correctly.

-Are there concerns about ethical or regulatory requirements being met? The only ethical concern is the ethical or review board approval for access to medical surveillance data. No mention was made as to whether the data was de-identified prior to supply of data or not. It likely was but this should be made clear.

Reviewer #2: Title:

1.The title does not perfectly represent the statistical analysis performed in this study. The statistical analysis was partly covered for temporal analysis. I do suggest the author improved the title and correct writing of the full name of hemorrhagic fever with renal syndrome. 

Background:

1.It is recommended to add references to the prediction model of HFRS and analysis of related factors in 2022.

Material & Methods:

1.Authors have to show ethical approve (approval No.) about patient’s specimens. Also, It is unclear by what criteria the clinical HFRS case was classified and lab diagnose was conducted (the ‘Diagnose criteria’ in ‘Methods’ section is not enough to explain). More detailed explanations were needed.

**Results**

-Does the analysis presented match the analysis plan?

-Are the results clearly and completely presented?

-Are the figures (Tables, Images) of sufficient quality for clarity?

Reviewer #1: Yes this is adequately represented in the clear tables and graphs supplied.

Reviewer #2: Results:

1.Please explain the division of seasons in Table S1.

2.Can some graphic parameters in Figure S9 be further optimized?

3.Result 3.4 indicate that high levels of SO2 can have a transient or continuous effect on HFRS, with the effect initially concentrated in the lag time of 0-5 days and after 15 days. The effect of high SO2 concentration on the disease is not very clear, so the conclusion that SO2 in the range of 200-250 μg/m3 increases the risk of HFRS infection and is significant in women and patients with delayed onset of 5-9 days is questionable.

4.It is necessary to indicate the specific chart in which Result 3.5 is presented.

5.It is recommended to highlight meaningful research results in Figure 3.

**Conclusions**

-Are the conclusions supported by the data presented?

-Are the limitations of analysis clearly described?

-Do the authors discuss how these data can be helpful to advance our understanding of the topic under study?

-Is public health relevance addressed?

Reviewer #1: Yes, this study shows the association of air pollution and meteorological variables with HFRS incidence in Sheyang in China. The study limitations are clearly stated and the authors do show how these data can assist with understanding the interactions between the environment and infectious disease dynamics.

Reviewer #2: Conclusion:

Does not represent at all from the result and discussion. With only three lines, i don't see it fits to be a conclusion.

**Editorial and Data Presentation Modifications?**

Reviewer #1: The study would benefit from the added use of other climatic variables to strengthen the potential links with climate.

Reviewer #2: (No Response)

**Summary and General Comments**

Reviewer #1: There are some typographical errors.

Abstract - there are spaces between SO and 2 and also 12 as a subscript is missing.

Introduction - last 2 paragraphs

J. Roda Gracia reference should have the year in parenthesis.

All references should have a space between the last word and reference.

Boosted regression tree (BRT) 'verifing' should be 'verifying'.

Discussion -

3rd paragraph 'Zhanget al' and 'Zhaoet al' should be 'Zhang et al' and 'Zhao et al'

4th paragraph 'Liet al. 2021' should be 'Li et al. 2021'.

Reviewer #2: (No Response)

PLOS authors have the option to publish the peer review history of their article (what does this mean?). If published, this will include your full peer review and any attached files.

Reviewer #1: No

Reviewer #2: No

Figure Files:

Data Requirements:

Reproducibility:

References

---

## [Decision Letter · Decision Letter 1]

4 May 2023

Dear Dr Dong,

Thank you very much for submitting your manuscript "Time series analyses based on the joint lagged effect analysis of pollution and meteorological factors of HFRS and the construction of prediction model" for consideration at PLOS Neglected Tropical Diseases. As with all papers reviewed by the journal, your manuscript has been reviewed by members of the editorial board and by independent reviewers. 

We are likely to accept this manuscript for publication, providing that you modify the manuscript according to our editorial recommendations below this letter. Most of the remaining issues are due to inadequate or missing responses to the input from the first review.

Important additional instructions are given below the editorial comments. 

Sincerely,

Christopher M. Barker

Academic Editor

Victoria Brookes

Section Editor

Editorial input:

Title: Spell out "hemorrhagic fever with renal syndrome" as suggested in the first review.

Throughout the manuscript, as pointed out in the first review, in-text citation numbers must be preceded by a space as required by PNTD guidelines. This was not corrected.

lines 111-112: as before, this statement seems to compare cases over a whole decade (1980s) to one year. Please correct to compare annual averages. A single year does not imply a trend. Can you not point instead to the range of cases from 2000-2009, similar to the data for the 1980s?

lines 119-120: Climatic conditions are plural. Change to "Climatic conditions are broadly regarded as some of the most pivotal factors affecting rodent population dynamics..."

lines 138-143: The need for so many different modeling approaches remains weakly justified, but I will accept this as written. Names of models should be plural and should not be capitalized in this paragraph (e.g., boosted regression trees, distributed lag nonlinear models, generalized additive models) 

lines 161-163: Which Ministry of Health? Local Shenyang MoH, or all of China? Are these guidelines not published? If they are, please cite the website or other source.

lines 168-179: This entire section needs rewording for clarity. There are problems with redundancy (e.g., "diagnostic cases of clinical diagnosis cases") and also many issues with sentence structure. Colons (:) are used, apparently to separate diagnostic criteria, and the entire section is included as if it were a single sentence (no periods are found in the 12-line section). Also, it is impossible to determine the case definition from the long list of criteria - is every condition listed required in order to be a clinically diagnosed case? The minimum requirements to be counted as a diagnosed case must be clear.

line 181: As stated previously, (www.data.cma.cn) needs a proper website citation in the bibliography. Websites are cited as references, not simple URLs in the text. See style #38 here: https://www.nlm.nih.gov/bsd/uniform_requirements.html

lines 263-264: As stated previously, the incubation period (lag from infection to disease onset) is not the only consideration here. What about the lag from environmental factors to changes in vulnerability of the human populations to infection? The response to the comment did not answer the question asked. I understand that the model considered a delay of up to 16 days based on QAIC, but why were longer lags not considered that would account for the lagged effects of the environment on vulnerability to infection and disease followed by the incubation period?

lines 268-269: Sentence remains unclear. What is meant by "comparing the 25th or 75th

269 percentile of environmental variables"? Did you compare the 75th to the 25th percentile? Please reword for clarity.

lines 284-285: This clarification helps to know what was missing, but "sets of data" is still unclear. A data set can be quite large. Please specify exactly what percentage of data were missing for each of the indicated years and the length of the periods that were missing.

line 321: The phrase "modifying the model according to the model parameters" has no clear meaning. Please reword.

lines 321-324: Previous questions were not addressed in the authors' responses: Why would you base assessments only on a single year of validation data? Using a single year for validation is relatively weak. It would be better to repeat the exercise for all years, leaving each year out as the validation data, then using all other years for training. If the authors disagree, this needs an explanation in the text.

line 343: As stated in earlier input, Figure 5 is cited before Figures 3 and 4. This is out of order and must be fixed. The authors stated that they corrected this problem, but they did not fix all.

line 414: The word "Statistically" does not add meaning to this sentence and should be deleted.

Fig 4 caption: As suggested in the earlier review, the final phrase should be corrected to "...is used as a reference level."

Fig 5 caption: Please correct the wording. What is a "growth regression tree model"? This term is not used elsewhere in the paper.

Reviewer's Responses to Questions

Reviewer #1: The necessary changes have been made to the article that have improved the methodology and clarification provided for the choices of more than one method.

Figure Files:

Data Requirements:

Reproducibility:

References

---

## [Editor Report · Decision Letter 2]

26 Jun 2023

Dear Dr Dong,

We are pleased to inform you that your manuscript 'Time series analyses based on the joint lagged effect analysis of pollution and meteorological factors of hemorrhagic fever with renal syndrome and the construction of prediction model' has been provisionally accepted for publication in PLOS Neglected Tropical Diseases.

We have attached a Microsoft Word document, which is a copy of the "track changes" version of your revised manuscript with a few final comments and edits from us. We hope this helps to make the final changes easier, and we directly edited wordings where applicable.

Also, before your manuscript can be formally accepted you will need to complete some final formatting changes, which you will receive in a follow up email. A member of our team will be in touch with a set of requests.

Best regards,

Christopher M. Barker

Academic Editor

Victoria Brookes

Section Editor

---

## [Editor Report · Acceptance letter]

19 Jul 2023

Dear Dr Dong,

We are delighted to inform you that your manuscript, "Time series analyses based on the joint lagged effect analysis of pollution and meteorological factors of hemorrhagic fever with renal syndrome and the construction of prediction model," has been formally accepted for publication in PLOS Neglected Tropical Diseases.

Best regards,

Shaden Kamhawi

co-Editor-in-Chief

Paul Brindley

co-Editor-in-Chief
